# Fibroblast growth factor receptor 2 (*FGFR2*) genetic polymorphisms contribute to fused roots in human molars

Sandra Regina Santos Meyfarth[1], Flares Baratto-Filho[2,3], Maria Eduarda Nunis Locks[2], Peter Proff[4], Giordano Oliveira Zandoná[3], Thaís de Oliveira Fernandes[5], Paulo Henrique Condeixa de França [2], Christian Kirschneck[6], Leonardo Santos Antunes[7], Erika Calvano Küchler [6]*

**1** Postgraduation Program, School of Dentistry, Fluminense Federal University, Niterói, RJ, Brazil, **2** Department of Dentistry, University of Joinville Region (Univille), Joinville, South Carolina, Brazil, **3** School of Dentistry, Tuiuti University from Paraná, Curitiba, PR, Brazil, **4** Department of Orthodontics, University of Regensburg, Regensburg, Germany, **5** Postgraduation Program, Fluminense Federal University, Nova Friburgo Health Institute, Nova Friburgo, RJ, Brazil, **6** Department of Orthodontics, University Hospital of Bonn, Bonn, Germany, **7** Department of Specific Formation, Fluminense Federal University, Niterói, RJ, Brazil

* erikacalvano@gmail.com

## Abstract

Fibroblast growth factors (FGFRs) signaling are required for human tooth development. Its dysregulation affects tooth formation and patients with *FGFR2* mutations often present dental anomalies in the spectrum of the syndrome. This study aimed to investigate whether genetic polymorphisms in *FGFR2* are associated with molar fused roots. The null hypothesis is that genetic variations in *FGFR2* are not associated with isolated cases (non-syndromic) of molars fused roots. Panoramic radiographs of non-syndromic patients were used to assess the occurrence of fused roots in molars. Genomic DNA analysis was performed to investigate polymorphisms within the candidate gene. The association between fused roots and genetic polymorphisms was analyzed using allelic and genotypic distributions, and haplotype frequencies. Odds ratio and 95% confidence interval were calculated to assess the chance of presenting fused roots. The significance level was set at $p < 0.05$ for all the analysis. A total of 170 patients were included. Statistically significant differences in genotype distribution were observed in rs10736303 and rs2162540. Individuals carrying at least one G allele of rs10736303 had an increased chance to present fused roots. A total of 154 haplotype combinations demonstrated statistically significant associations. The polymorphisms rs10736303 and rs2162540 in *FGFR2* were associated with fused roots in human molars.

## Introduction

Fused roots in molars are a morphological dental anomaly where two or more roots become merges due to disruptions in the development of the Hertwig epithelial root sheath [1,2],

**Data availability statement:** All data is available as supplementary material.

**Funding:** This study was financed in part by the Alexander-von-Humboldt Foundation and Coordenação de Aperfeiçoamento de Pessoal de Nível Superior - Brasil (CAPES) - Finance code 001. This work was supported by the Open Access Publication Fund of the University of Bonn.

**Competing interests:** The authors have declared that no competing interests exist.

which starts as a bilayered extension of the inner and outer dental epithelium, originating from the cervical loop of the enamel organ. This epithelial double layer extends apically, shaping the future root [3].

Proper root development is essential for effective dental function and longevity, and anomalies such fused roots, can pose challenges in dental treatments. In endodontics, fused roots often have complex canal systems with additional grooves and connections making cleaning, shaping, and filling more difficult, and requiring careful management to avoid perforations [4,5]. Prosthetically, fused roots result in a smaller root surface area and a less favorable crown-root ratio, leading to reduced stability [6]. In Periodontics, fused roots can lead to bacterial migration due to grooves on the root surface, impairing periodontal tissue resistance and increasing the risk of bone destruction [6–8].

Tooth root development involves complex cellular processes and molecular controls, with Hertwig's epithelial root sheath (HERS) directing odontoblast differentiation and radicular dentin formation. Signaling pathways, which must be tightly regulated, influence the activities of HERS and the differentiation of dental mesenchymal cells [9]. Disruptions in these processes due to genetic variations, can lead to dental anomalies such as variations in tooth number, shape, eruption, or hard tissue formation [9]. These anomalies are often observed as isolated that only the dentition is affected, like tooth agenesis and hypodontia being particularly prevalent. Additionally, various rare developmental syndromes display dental anomalies, because the genes and genetic networks governing tooth formation frequently linked in the development of other organs and tissues. Key signaling molecules, such as fibroblast growth factors (FGFs) and their receptors, play crucial roles in root formation [10]. FGFs belong to the heparin-binding growth factor superfamily and are essential for early embryonic development, including the dentition and act through specific receptors (FGFRs) [11].

The identification of genes associated with rare syndromes offers valuable insights into understanding more prevalent isolated traits. In recent years, genes linked to rare autosomal syndromes, have been contributing significantly to our knowledge of facial anomalies and dental phenotypes [12]. *FGFR2* disorders span a spectrum from some specific syndromes, such as Crouzon, Apert, and Pfeiffer, to the dental alterations [13], skeletal malocclusions [14], and craniofacial malformation [15]. In syndromic patients with *FGFR2* mutations, cases of dental root fusion have been observed [16–23]. Despite growing interest in the genetic and developmental aspects of root formation, this area remains underexplored [24]. The dental phenotypes observed in patients with *FGFR2* disorders and the crucial role of FGFs in tooth development may indicate that genetic variations in *FGFR2* also contribute to isolated cases (non-syndromic) of dental root fusion. The hypothesis of this study is that variations in *FGFR2* are involved with fused roots of non-syndromic patients. Therefore, we aimed to investigate whether genetic polymorphisms in *FGFR2* are associated with fused roots in molars.

## Materials and methods

### Ethical aspects and type of study

This study received the approval of the Local Research Ethics committee of Regensburg University, Germany (# 19-1549-101) and were performed in accordance with the latest version of Helsinki Declaration guidelines. Informed consent was obtained from all the participants. This is a nested cross-sectional phenotype-genotype association study that used the Strengthening the Reporting of Genetic Association study (STREGA) statement checklist [25] to design and conduct the study.

## Subjects and sample calculation

Orthodontic patients aged from 16 to 47 years old presenting digital panoramic radiographs were recruited from the same institution, University of Regensburg, Germany between 2020 and 2021 in an attempt to select cases and control individuals with similar ethnicity, and social-culture backgrounds.

GPower software (Franz Faul University, Kiel, Germany) was used for sample size calculation with a power of 80% and alpha of 5% using the data found in Ross & Evanchik [26], in which observed a frequency of 29% of fused root in all molars. Therefore, the sample size was calculated in the case to control ratio of 1:3 assuming a 25% difference of the genotype's frequencies between groups. The estimation for the total sample of 156 patients were required (case to control ratio 39:117). Therefore, we recruited a convenience sample of consecutively orthodontic patients.

## Inclusion/exclusion criteria

This study included patients with European ancestry (self-reported ethnicity). Only orthodontic patients presenting high quality panoramic radiographs, without any extraction of the upper and lower first and second molars, with no tooth loss due to carious lesions or trauma, and without agenesis of permanent teeth were included. Subjects who presented syndromes identified previously in the anamnesis, oral cleft, low-quality radiographs (e.g., poor angulation, improper exposure, or faulty processing), teeth with extensive root resorption processes, and immature roots were excluded from the analysis. Third molars were not considered, since the third molar roots are not completely formed at the initial age of recruitment of the majority of the included patients and due the fact that this anatomical region is more difficult to visualize.

## Evaluation of fused roots phenotype

Briefly, a dentist (MENL) not involved in the genotyping analysis, blindly performed the phenotypic analyses, after training and calibration with a senior dentist (FBF) using the same protocol. Inter-observer (0.81) and intra-observer (0.94) concordance were assessed by Cohen's Kappa, with a very good agreement. All the digital panoramic radiographs with high-quality requirements were examined digitally in a dark room using the Windows viewer software for Windows 10 (Microsoft Corporation, Redmond, WA, USA) on a 14-in Lenovo 81V7S00100 monitor (Lenovo PC International, Beijing, China) with a resolution of 1,360 × 768 pixels.

Each permanent molar tooth (excluding third molars) was radiographically evaluated. Teeth presenting roots fused apically beyond the usual furcal position, lacking radiographic indications of periodontal ligament space, and exhibiting the absence of bone between the distinct roots of the molars at any apical level to the bifurcation [26], as demonstrated in Fig 1, were considered with fused roots. This classification applied to molars with fusion involving one-third or less of their roots, as well as those with fusion encompassing the entire root surfaces. The fusion's location did not influence this classification; it could occur in the apical, middle, or cervical third, or in any combination of these sections. In some instances, molars had roots fused only in the apical one-third, while maintaining a normal furcation with intact alveolar bone and other periodontal structures. These cases were still categorized as having fused roots to ensure clarity and consistency. Based on these radiographic parameters, the upper and lower first and second molars were assessed for presence or absence of root fusion, and the patients included in case (at least one molar with fused root) and control (all molars without fused roots) groups.

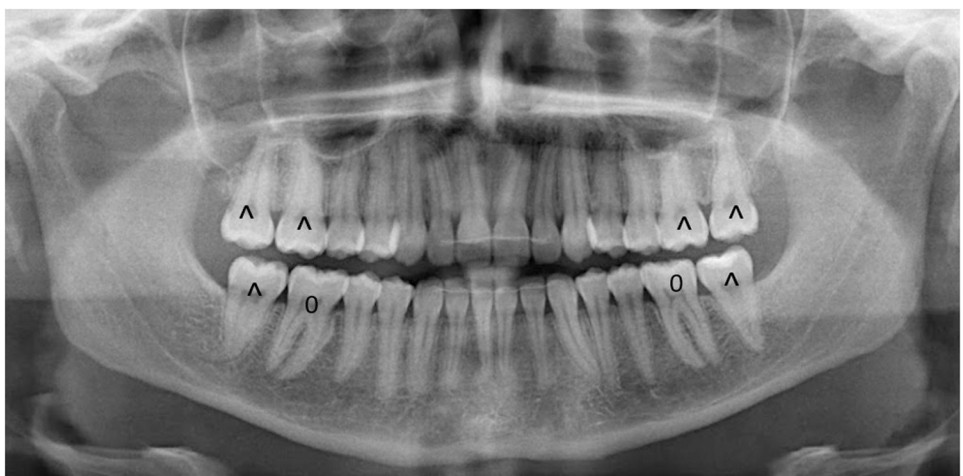

**Fig 1. Panoramic radiographs showing molars with and without fused roots.** Note: ^ means fused roots; 0 no fused roots.

### Electronic search for the candidate gene selection

For the selection of the candidate gene, the first step was an investigation through a brief literature review to retrieve cases of patients with *FGFR2* mutations and root fusion. The studies should present radiographs of patients for the analysis. An electronic search was conducted in the PubMed database using a combination of MeSH and free terms: Receptor Fibroblast Growth Factor type 2; syndromes; dental morphology; tooth abnormalities; tooth eruption; dental anomalies.

The studies retrieved in the literature revealed that syndromes associated with mutations in the *FGFR2* gene often manifest fused roots in molars as seen in the radiographs of the published cases. Some of the root phenotypes observed in the studies of the patients with *FGFR2* mutations [16–23] are presented in Fig 2, suggesting that *FGFR2* plays an important role in dental root development and is a candidate gene for non-syndromic molars fused roots.

### Selection of the studied genetic polymorphisms

The genetic polymorphisms in *FGFR2* were selected based on their minor allele frequency (higher than 20%) and their previous association with dentofacial phenotypes [27,28]. The description of the selected genetic polymorphisms is presented in the Table 1. Six genetic polymorphisms are located in intron 2 of *FGFR2*, which probably plays a vital role in the expression of *FGFR2* [14].

### DNA extraction

Genomic DNA was extracted from cells isolated from saliva samples collected from all participants as a previous protocol established by Küchler et al. [29]. Briefly, a sample of cheek cells was collected using cytobrushes and stored in 1 ml of extraction buffer (TE) (10 mM Tris HCl, pH 7.8; 5 mM EDTA; 0.5% SDS) at -20°C until processing.

The samples were defrosted and incubated with 100 ng/ml of Proteinase K in a water bath at 56°C overnight and subjected to precipitation processes using 400 μL of 10 M ammonium acetate solution. Then, all tubes were shaken for 5 min and centrifuged for 15 min (12,000 rpm). The supernatant was divided into two tubes of 700 μL each. The same volume of ice-cold

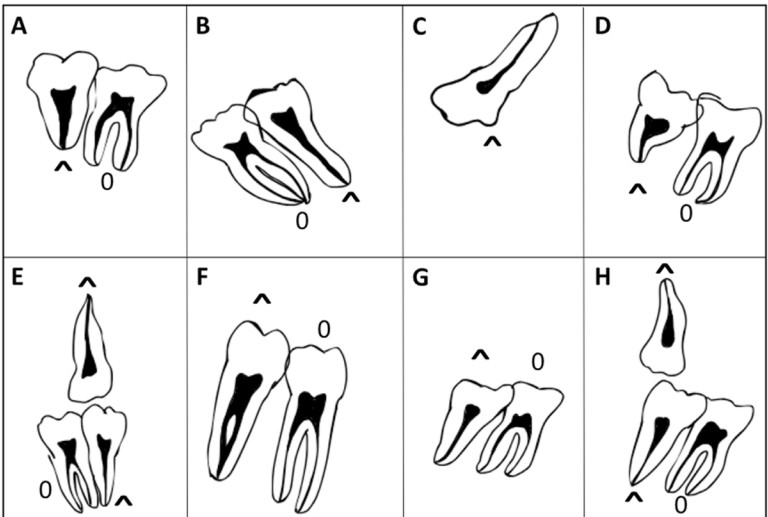

**Fig 2. Schematic summarizing morphological variations of fused roots found from literature on patients presenting mutation in *FGFR2*.** a: Horiuchi et al., 2021 [16]; b: Costa et al., 2012 [17]; c: Droubi et al., 2022 [18]; d: Hassona et al., 2017 [19]; e: Nagaraju et al., 2016 [20]; f: Park et al., 2011 [21]; g: Saberi et al., 2011 [22]; h: Tripathi et al., 2021 [23].

Table 1. Description of the selected genetic polymorphisms.

| Chromosome position | rs | Assay ID[b] | Base[b] change | MAF[a] | Location and description |
|---|---|---|---|---|---|
| 10:121592622 | rs2162540 | C___2917304_10 | C > T | 0.433 | Located on Intron 2; associated with skeletal malocclusion [14,35] |
| 10:121580797 | rs2981578 | C___2917323_20 | C > T | 0.394 | Located on Intron 2; affect bone mineral density and associated with skeletal III malocclusions [14,37] |
| 10:121574943 | rs10736303 | C___2917332_10 | A > G | 0.392 | Located on Intron 2; affect bone mineral density and skeletal class III malocclusions [14,35,37,46] |
| 10:121575416 | rs11200014 | C__31019228_10 | A > G | 0.341 | Located on Intron 2; associated with skeletal classes II and III malocclusions [14,35] |
| 10:121586676 | rs1219648 | C___2917314_20 | A > G | 0.395 | Located on Intron 2 variant; associated with oral cleft and tooth agenesis [27] |
| 10:121508117 | rs4752566 | C__11309749_10 | G > T | 0.374 | Located on Intron 9 variant; associated with oral cleft [28] |
| 10:121579461 | rs1078806 | C___8899701_10 | A > G | 0.340 | Located on Intron 2; affect bone mineral density [37] |

Note:

[a]MAF means minor allele frequency. Data obtained from databases:

[b]http://www.thermofisher.com;

[b]https://www.ncbi.nlm.nih.gov/snp/rs4803455.

isopropyl alcohol (700 μL) was added to each specimen and was manually and vigorously shaken, and then centrifuged for 20 min at 12,000 rpm. The supernatant was discarded and 1 ml of ice-cold 70% ethanol was added and centrifuged for 15 min at 12,000 rpm. The supernatant was discarded and dried. The DNA pellet was resuspended in 50 μL of TE and frozen at -20°C.

## DNA evaluation and genotype analysis

The assessment of DNA quantity and purity was conducted using spectrophotometry (Nano-drop 1000; Thermo Scientific) using 2 μL of the extracted material. The DNA concentration was assessed at a wavelength of 260 nm. The ratio between the values obtained at wavelengths of 260 nm and 280 nm was used to estimate the purity of genomic DNA. Only DNA samples with a 260/280 ratio between 1.6 to 2.0 were included in this study.

The method used for genotyping was TaqMan with specific probes for allelic distinction (TaqMan probes). The real-time PCR reaction was performed after the reaction was optimized for the experimental conditions, in which a final volume of 3 µL (4 ng of DNA/reaction, 1.5 µL of TaqMan PCR master mix, 0.125 µL of SNP assay-byDesign (Applied Biosystems-Foster City, CA) and deionized water q.s.p.) was used. For amplification, 40 cycles were performed at 95°C for 10 minutes, 92°C for 15 seconds and 60°C for 1 min, as established by Ranade et al. [30]. The polymorphisms were genotyped blindly via real-time polymerase chain reaction (PCR) utilizing the TaqMan assay (Step One Plus Real-Time PCR System, Applied Biosystems, Foster City, CA). To confirm the accuracy of genotyping, 10% of the samples were randomly selected for repeated genotyping by real time PCR and the result consistency was 100%.

## Statistical analysis

All the analysis were performed using PLINK software version 1.06 (https://zzz.bwh.harvard.edu/plink/ld).

Chi-square test was used to compare the genotype and allelic distributions between control and fused root groups, and haplotype frequencies.

The genotype analysis was performed in the co dominant (aa vs. aA vs. AA), dominant (aa + aA vs. AA), and recessive models (aa vs. aA + AA). Odds ratios (OR) and 95% confidence intervals (CI) were calculated to evaluate the chance of presenting fused roots in the allelic discrimination analysis.

The Hardy-Weinberg equilibrium (HWE) was also evaluated by Chi-square test.

In the analysis of each genetic polymorphisms, cases with missing values were dropped from the analysis. All tests were performed with a significance level set at $p < 0.05$.

## Results

### Characteristics of the included sample

Of the 175 patients initially screened for the study, 5 of them were excluded due to low quality of the radiographs. Finally, the orthodontic digital panoramic radiographs and genomic DNA of 170 patients (86 females and 84 males), and 1360 molars were analyzed. Of the total patients, 124 did not have any molar with fused roots, and 46 had at least one molar with fused roots. The most common affected tooth was upper second molars (n = 56) (Table 2).

**Table 2. Demographic characteristics of the sample.**

| Age | 16-47 years old |
|---|---|
| Gender | n |
| Male | 84 |
| Female | 86 |
| Molars analyses | n |
| Total molars evaluated | 1360 |
| Molars without fused roots | 124 |
| At least one molar fused root | 46 |
| First upper molars fused roots | 38 |
| First lower molars fused roots | 0 |
| Second upper molars fused roots | 56 |
| Second lower molars fused roots | 19 |

## Hardy-Weinberg equilibrium results

All the genetic polymorphisms assessed were within the HWE ($\chi 2$value for rs4752566 = 1; $\chi 2$value for rs10736303 = 0.34; $\chi 2$value for rs11200014 = 1; $\chi 2$value for rs1078806 = 0.75; $\chi 2$value for rs2981578 = 0.75; $\chi 2$value for rs1219648 = 0.31; $\chi 2$value for rs2162540 = 0.32).

## Genotype results

Table 3 shows the genotype distributions among fused roots and control group. The rs10736303 in *FGFR2* showed a statistically significant difference in the co-dominant (p

**Table 3. Genotype distribution between genetic polymorphisms in *FGFR2*.**

| rs | Phenotype | Genotype (%) | | | p-value | | |
|---|---|---|---|---|---|---|---|
| | | TT | TG | GG | *Co-dominant* | *Dominant* | *Recessive* |
| rs4752566 | Control (without fusion) | 16 | 60 | 37 | *reference* | | |
| | Molar (maxillary + mandibular) with fused root | 7 | 21 | 13 | 0.904 | 0.903 | 0.653 |
| | Maxillary molar with fused root | 6 | 20 | 12 | 0.968 | 0.894 | 0.805 |
| | Mandibular molar with fused root | 2 | 6 | 3 | 0.900 | 0.711 | 0.717 |
| rs10736303 | | GG | GA | AA | *Co-dominant* | *Dominant* | *Recessive* |
| | Control (without fusion) | 10 | 71 | 36 | *reference* | | |
| | Molar (maxillary + mandibular) with fused root | 13 | 17 | 11 | 0.0012* | 0.634 | 0.0002* |
| | Maxillary molar with fused root | 10 | 17 | 11 | 0.015* | 0.831 | 0.004* |
| | Mandibular molar with fused root | 5 | 4 | 2 | 0.0013* | 0.382 | 0.0002* |
| rs11200014 | | AA | AG | GG | *Co-dominant* | *Dominant* | *Recessive* |
| | Control (without fusion) | 10 | 58 | 44 | *reference* | | |
| | Molar (maxillary + mandibular) with fused root | 6 | 18 | 15 | 0.515 | 0.927 | 0.259 |
| | Maxillary molar with fused root | 5 | 17 | 14 | 0.678 | 0.966 | 0.391 |
| | Mandibular molar with fused root | 2 | 4 | 5 | 0.485 | 0.69 | 0.323 |
| rs1078806 | | GG | GA | AA | *Co-dominant* | *Dominant* | *Recessive* |
| | Control (without fusion) | 17 | 58 | 42 | *reference* | | |
| | Molar (maxillary + mandibular) with fused root | 8 | 19 | 15 | 0.770 | 0.983 | 0.490 |
| | Maxillary molar with fused root | 7 | 17 | 15 | 0.781 | 0.773 | 0.608 |
| | Mandibular molar with fused root | 2 | 5 | 4 | 0.939 | 0.975 | 0.744 |
| rs2981578 | | TT | TC | CC | *Co-dominant* | *Dominant* | *Recessive* |
| | Control (without fusion) | 22 | 64 | 23 | *reference* | | |
| | Molar (maxillary + mandibular) with fused root | 8 | 23 | 11 | 0.798 | 0.502 | 0.875 |
| | Maxillary molar with fused root | 8 | 21 | 10 | 0.824 | 0.558 | 0.965 |
| | Mandibular molar with fused root | 2 | 6 | 3 | 0.893 | 0.635 | 0.874 |
| rs1219648 | | GG | GA | AA | *Co-dominant* | *Dominant* | *Recessive* |
| | Control (without fusion) | 13 | 69 | 44 | *reference* | | |
| | Molar (maxillary + mandibular) with fused root | 7 | 16 | 19 | 0.333 | 0.407 | 0.361 |
| | Maxillary molar with fused root | 7 | 15 | 17 | 0.331 | 0.531 | 0.277 |
| | Mandibular molar with fused root | 1 | 3 | 7 | 0.241 | 0.096 | 0.830 |
| rs2162540 | | CC | CT | TT | *Co-dominant* | *Dominant* | *Recessive* |
| | Control (without fusion) | 13 | 76 | 33 | *reference* | | |
| | Molar (maxillary + mandibular) with fused root | 7 | 17 | 20 | 0.024* | 0.024* | 0.358 |
| | Maxillary molar with fused root | 7 | 15 | 19 | 0.016* | 0.021* | 0.278 |
| | Mandibular molar with fused root | 1 | 3 | 7 | 0.036* | 0.011* | 0.871 |

Note: All comparisons were performed with the control group that was used as reference for all analysis. * Means statistical significance.

= 0.0012; p = 0.015; p = 0.0013) and recessive models (p = 0.0002; p =0.004; p = 0.0002) for molars (maxillary + mandibular), maxillary molars, and mandibular molars with fused roots, respectively. Also, the rs2162540 showed a significant difference in the co-dominant (p = 0.024; p = 0.016; p = 0.036) and dominant models (p = 0.024; p = 0.021; p = 0.011) for molars (maxillary + mandibular), maxillary molars, and mandibular molars with fused roots, respectively.

## Allelic distribution results

Individuals carrying at least one G allele in the rs10736303 had an increased chance to present fused roots in molars (maxillary + mandibular) (OR = 1.73, CI 95% 1.04-2.87; p = 0.032), and in mandibular molars (OR = 2.75, CI 95% 1.11-6.81; p = 0.024) compared to controls (Table 4).

**Table 4. P-value, odds ratio and confidence interval for the allelic distribution of *FGFR2* among groups.**

| rs | Phenotype | Minor Allele Frequency (%) | p-value | OR (95% CI) |
|---|---|---|---|---|
| rs4752566 (Minor allele T) | Control (without fusion) | 40.71 | *reference* | *reference* |
| | Molar (maxillary + mandibular) with fused root | 42.68 | 0.755 | 1.08 (0.65 - 1.80) |
| | Maxillary molar with fused root | 42.11 | 0.830 | 1.05 (0.62 - 1.79) |
| | Mandibular molar with fused root | 45.45 | 0.665 | 1.21 (0.50 - 2.92) |
| rs10736303 (Minor allele G) | Control (without fusion) | 38.89 | *reference* | *reference* |
| | Molar (maxillary + mandibular) with fused root | 52.44 | 0.032* | 1.73 (1.04 - 2.87) |
| | Maxillary molar with fused root | 48.68 | 0.131 | 1.49 (0.88 - 2.51) |
| | Mandibular molar with fused root | 63.64 | 0.024* | 2.75 (1.11 - 6.81) |
| rs11200014 (Minor allele A) | Control (without fusion) | 34.82 | *reference* | *reference* |
| | Molar (maxillary + mandibular) with fused root | 38.46 | 0.563 | 1.17 (0.68 - 1.99) |
| | Maxillary molar with fused root | 37.50 | 0.679 | 1.12 (0.64 - 1.94) |
| | Mandibular molar with fused root | 36.36 | 0.884 | 1.07 (0.43 - 2.66) |
| rs1078806 (Minor allele G) | Control (without fusion) | 39.32 | *reference* | *reference* |
| | Molar (maxillary + mandibular) with fused root | 41.67 | 0.705 | 1.10 (0.66 - 1.83) |
| | Maxillary molar with fused root | 39.74 | 0.946 | 1.01 (0.60 - 1.71) |
| | Mandibular molar with fused root | 40.91 | 0.883 | 1.06 (0.43 - 2.60) |
| rs2981578 (Minor allele T) | Control (without fusion) | 49.54 | *reference* | *reference* |
| | Molar (maxillary + mandibular) with fused root | 46.43 | 0.627 | 0.88 (0.53 - 1.46) |
| | Maxillary molar with fused root | 47.44 | 0.749 | 0.91 (0.54 - 1.54) |
| | Mandibular molar with fused root | 45.45 | 0.714 | 0.84 (0.35 - 2.04) |
| rs1219648 (Minor allele G) | Control (without fusion) | 36.64 | *reference* | *reference* |
| | Molar (maxillary + mandibular) with fused root | 35.71 | 0.880 | 0.96 (0.57 - 1.61) |
| | Maxillary molar with fused root | 37.18 | 0.931 | 1.02 (0.60 - 1.74) |
| | Mandibular molar with fused root | 22.73 | 0.192 | 0.50 (0.18 - 1.42) |
| rs2162540 (Minor allele C) | Control (without fusion) | 41.80 | *reference* | *reference* |
| | Molar (maxillary + mandibular) with fused root | 35.23 | 0.280 | 0.75 (0.45 - 1.25) |
| | Maxillary molar with fused root | 35.37 | 0.303 | 0.76 (0.45 - 1.28) |
| | Mandibular molar with fused root | 22.73 | 0.08 | 0.40 (0.14 - 1.14) |

Note: All comparisons were performed with the control group that was used as reference for all analysis. OR = Odds ratio; CI = Confidence Interval; * means statistical significance.

## Haplotype results

Haplotypes of the 7 polymorphisms were analyzed. The haplotypes frequency comparisons are presented in the S1 Table. A total of 154 combinations of haplotypes showed statistically significant value ($p < 0.05$).

## Discussion

This study found a significant association between genetic polymorphisms in *FGFR2* and molars fused roots in non-syndromic individuals. Therefore, the hypothesis supported that genetic polymorphisms in *FGFR2* are associated with fused roots in molars.

FGFR2 is a highly conserved receptor tyrosine kinase [31] upstream of several signal transduction pathways, such as RAS/mitogen-activating protein (MAP) kinase, the phosphoinositide 3 (PI3) kinase/ AKT, and the phospholipase C gamma (PLC $_\gamma$) [32], that are crucial for osteogenic differentiation [33]. FGF signaling serves multiple essential functions during embryo development [34], tooth morphogenesis, cellular proliferation, differentiation, and migration, and patterning. FGF-FGFR signaling is also critical to the developing axial and craniofacial skeleton [32]. It is noteworthy that the selected genetic polymorphisms in *FGFR2* were situated within intron 2 (excepting rs4752566) of long *FGFR2* isoform transcripts and exhibited close linkage. These genetic polymorphisms located in intron 2 of *FGFR2* have been reported as associated with skeletal malocclusion, osteoporosis and breast cancer in previous studies [14,35–37].

Moreover, there exists a potential active enhancer region within the intron 2 of *FGFR2*, coinciding with the positions of rs298157816 and rs10736303. This suggests that the second intron of *FGFR2* likely holds significance in *FGFR2* expression. Investigating the transcriptional regulation mechanism of intron 2 could offer insights into the underlying causes of *FGFR2*-associated phenotypes, including root developmental alterations. It is notable that the same genetic locus appears to influence phenotypes ranging from extremely rare syndromic forms of craniofacial malformations, such as Crouzon, Apert, and Pfeiffer [12,19,38], to the highly common dental anomalies as tooth agenesis and fused root phenotypes, suggesting a continuum within the same clinical spectrum as it was possible to observe in the studies retrieved in the literature [16,17,19–23].

The results of this study revealed an association between rs10736303, rs2162540, and root fusions in molars. The associated genetic polymorphism located within intron 2, such as rs10736303, was discovered to serve as binding site for *RUNX2* and *SMAD4* [39,40]. The interaction between *RUNX2* and *SMAD4* was observed to enhance the expression of *FGFR2*. The rs10736303 contained the binding sites of *RUNX2* and *SMAD4* and the polymorphism may enhance the effect of *RUNX2* and *SMAD4*, as well the levels of *FGFR2* expression [14]. Interestingly, rs10736303 was associated with fused roots in our study. FGF/FGFR signaling induces the expression of *RUNX2*, which is a key transcription factor in osteoblast differentiation [41]. It is worth noting that *RUNX2* act as pivotal regulators in the initial signaling pathways regulating crucial epithelial-mesenchymal [42]. The epithelial mesenchymal interactions regulate all aspects of tooth development, including initiation, and morphogenesis of root formation. Notably, a study performed by Merametdjian et al. [43], found several dental anomalies in patients with *RUNX2* mutation, such as hypodontia, microdontia, taurodontism, radiculomegaly. *SMAD4* plays a crucial role in regulating tooth root development [44]. Studies in mice showed that ablation of Smad4 in odontoblasts using Osteocalcin-Cre leads to shortened roots, impaired odontoblast differentiation, and the formation of osteodentin [45,46]. This variation rs10736303 was significantly associated with skeletal class II malocclusion [14]. Future studies should also investigate polymorphisms in *RUNX2* and *SMAD4*. Regarding rs2162540 in *FGFR2,* it was also associated with molar fused roots. This genetic polymorphism has been

identified in other studies as susceptibility variant also for skeletal malocclusions [14,35,47] demonstrating the importance of these polymorphisms for craniofacial development.

Although fused roots are relatively common in dental practice, the studies about tooth roots have been relatively sparse in dental research, compared to tooth crowns. However, advancements in technology have expanded the possibilities for analyzing variation in root morphology, thereby enhancing their significance as phenotypes. This study introduces a novel data regarding the role of genetic polymorphisms and fused roots. It is likely that other genes involved in root development may also be involved in the etiology of fused roots.

This study has some limitations. The use of digital panoramic radiographs for identifying molars fused roots may not always capture this phenotype accurately in a two-dimensional image. Consequently, the number of fused roots may have been underestimated. To mitigate this limitation, third molars were excluded from the analysis due to the challenges associated with assessing root morphology in this type of tooth [48].

The multiple comparisons in this study can be a limitation, as they increase the likelihood of a Type I error. However, we chose not to apply any multiple comparison adjustments, because such adjustments may lead to interpretative errors when the data being evaluated represent actual observations rather than random numbers, potentially increasing the risk of a Type II error. According to Perneger [49], adjusting statistical significance based on the number of tests performed on study data can create more issues than it resolves. The primary drawback is that the interpretation of a result becomes dependent on the total number of tests conducted.

Despite the limitations associated with the two-dimensional approach to phenotype determination, this study provides valuable insights into the potential involvement of *FGFR2* and fused roots phenotype in human molars. Understanding the genetic contribution to an increased chance of patients having fused roots and identifying the genes involved in this dental anomaly is a crucial step towards the development of future therapeutic approaches.

## Conclusions

The genetic polymorphisms rs10736303 and rs2162540 in *FGFR2* are associated with fused roots in molars. Patients presenting such variations could be involved in a higher chance to present molars with fused roots. Our study demonstrated that rare syndromes can act as models for understanding genetic susceptibility to more prevalent dental traits in the general population. Further investigations in different populations are needed to confirm these results.

## Supporting information

**S1 Table.  Haplotypes analysis of the SNPs in *FGFR2* statistically associated with molars fused roots.** Note: Bold indicates a statistically significant difference (p < 0.05). (DOCX)

## Acknowledgments

We are indebted to all participants. This work was supported by the Open Access Publication Fund of the University of Bonn.

## Author contributions

**Conceptualization:** Flares Baratto-Filho, Christian Kirschneck, Leonardo Santos Antunes, Erika Calvano Kuchler.

**Data curation:** Sandra Regina Santos Meyfarth, Flares Baratto-Filho, Christian Kirschneck, Leonardo Santos Antunes.

**Formal analysis:** Maria Eduarda Nunis Locks, Giordano Oliveira Zandoná, Thais de Oliveira Fernandes, Paulo Henrique Condeixa de França.

**Funding acquisition:** Christian Kirschneck, Erika Calvano Kuchler.

**Investigation:** Sandra Regina Santos Meyfarth, Flares Baratto-Filho, Paulo Henrique Condeixa de França, Erika Calvano Kuchler.

**Methodology:** Paulo Henrique Condeixa de França.

**Project administration:** Peter Proff, Christian Kirschneck, Erika Calvano Kuchler.

**Resources:** Peter Proff, Christian Kirschneck, Leonardo Santos Antunes, Erika Calvano Kuchler.

**Software:** Sandra Regina Santos Meyfarth.

**Supervision:** Flares Baratto-Filho, Christian Kirschneck, Leonardo Santos Antunes, Erika Calvano Kuchler.

**Validation:** Sandra Regina Santos Meyfarth, Flares Baratto-Filho.

**Visualization:** Peter Proff, Leonardo Santos Antunes.

**Writing – original draft:** Sandra Regina Santos Meyfarth.

**Writing – review & editing:** Flares Baratto-Filho, Maria Eduarda Nunis Locks, Peter Proff, Giordano Oliveira Zandoná, Thais de Oliveira Fernandes, Paulo Henrique Condeixa de França, Christian Kirschneck, Leonardo Santos Antunes, Erika Calvano Kuchler.

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
