## [Decision Letter · Decision Letter 0]

29 Jul 2024

PONE-D-24-26278Fibroblast growth factor receptor 2 (FGFR2) genetic polymorphisms contribute to fused roots in human molarsPLOS ONE

Dear Dr. Kuchler, Thank you for submitting your manuscript to PLOS ONE. After careful consideration, we feel that it has merit but does not fully meet PLOS ONE’s publication criteria as it currently stands. Therefore, we invite you to submit a revised version of the manuscript that addresses the points raised during the review process.

We look forward to receiving your revised manuscript.

Kind regards,

Sonam Khurana

Academic Editor

PLOS ONE

Journal Requirements:

We are indebted to all participants. This work was supported by the Open Access Publication Fund of the University of Bonn. This study was financed in part by the Alexander-von-Humboldt-Foundation. This study was financed in part by the Coordenação de Aperfeiçoamento de Pessoal de Nível Superior - Brasil (CAPES) - Finance Code 001.

"None"

Reviewers' comments:

Reviewer's Responses to Questions

**Comments to the Author**

1. Is the manuscript technically sound, and do the data support the conclusions?

Reviewer #1: Yes

Reviewer #2: Yes

Reviewer #3: Partly

Reviewer #4: Yes

Reviewer #5: Yes

Reviewer #6: Partly

2. Has the statistical analysis been performed appropriately and rigorously? 

Reviewer #1: Yes

Reviewer #2: Yes

Reviewer #3: I Don't Know

Reviewer #4: Yes

Reviewer #5: Yes

Reviewer #6: Yes

3. Have the authors made all data underlying the findings in their manuscript fully available?

Reviewer #1: Yes

Reviewer #2: Yes

Reviewer #3: Yes

Reviewer #4: Yes

Reviewer #5: Yes

Reviewer #6: Yes

4. Is the manuscript presented in an intelligible fashion and written in standard English?

Reviewer #1: Yes

Reviewer #2: Yes

Reviewer #3: Yes

Reviewer #4: Yes

Reviewer #5: Yes

Reviewer #6: Yes

5. Review Comments to the Author

**Reviewer #1:**  The study aims to explore whether genetic polymorphisms in FGFR2 are associated with fused roots in human molars, which is a valid research objective given the gene's known role in craniofacial development.However there's some points to clarify :

1- Ensure that the study has obtained appropriate ethical approvals and informed consent from participants. This is crucial for studies involving human subjects. as you didn't submit any ethics statement even it's included in the

Methods section of the manuscript.

2- The article reports associations between FGFR2 SNPs and fused roots but does not delve deeply into the biological mechanisms. Adding a discussion on how these genetic variations could affect dental development would strengthen the argument.

3 - Ensure the sample size calculation aligns with the study’s goals and statistical power. Confirm that the chosen sample size is adequate for detecting significant associations this will help to discuss the extent to which the findings can be generalized to other populations. If the sample is not representative of the broader population, this should be acknowledged.

**Reviewer #2:**  The authors conducted a good study about FGFR2 and his role efforts on the fused roots of molars. The study used good sample size and Investigative of FGFR2 in well method. Really, I enjoyed reading this article.

**Reviewer #3:**  There is interesting information presented in this paper and does contribute to the advancement of the field. Overall though, this paper is missing a developed discussion about why the authors chose to assess FGFR2, rather than a different gene, in conjunction with fused roots in humans. The absence of this information is notable in the introduction, discussion and conclusion sections. The presented hypothesis is also not clear and justification is missing. There is missing information in descriptions of sample selection and methods, and presentation of some statistical analyses. Sections of this paper could also be reorganized to provide better context and readability.

Abstract: The abstract is missing a reference to the hypothesis presented in the paper or specifics of the sample analyzed.

Line 50-52: What specific comparisons are the p-values referring to?

Line 54: The word "risk" is used in reference to OR. Since OR is a probability, this test is not assessing the risk, but rather than chance of having fused roots. Consider changing the wording here and elsewhere in the paper (see comments below).

Introduction: The introduction does not outline the point of the study very clearly or provide a clear hypothesis being tested.

Information in paragraph 2 (starting on line 67) could be better served as the leading paragraph to contextualize the study. Lines 62-66 could then be integrating into paragraph two of the introduction. While literature is mentioned in this introduction, no substantive background information or specific examples are provided.

Line 64-66: There are several stages of tooth development, how does this relate to root development specifically?

Paragraphs 7-9 from the discussion section (there are no line numbers in the discussion section, so paragraph numbers are used as a reference) provide some of the background information that would contextualize this study. I suggest integrating those paragraphs into the introduction and deleting them from the discussion section.

Line 79-82: Is this the hypothesis you are testing? There is no justification for why dental anomalies would be associated with non-syndromic cases. Please provide the background for this. In the subsequent line, this study sounds like an exploratory study rather than hypothesis testing. Please outline specific expectations with background justification if this is a hypothesis-testing study.

Materials and Methods: Sample breakdowns are not presented in the materials section. This is information is provided in the results section (168-172). Integrate that information within lines 92-101 and delete in the results section. Specific information about participant selection and identification of fused roots needs clarification.

Line 96: Define how you determined "case" and "control" and how many of each you have in this study.

Lines 102-108: You mention not using third molars. Specify which teeth you are analyzing for this study.

Line 103-106: Was there consideration to eruption stage or an age cut-off set to select participants? Please be specific about the criteria for participation.

Line 109-110: Which author(s) performed analyses? Please also indicate whether or not data collectors were blinded to group membership (the "case" and "control" groups). What protocol was used for identifying fused molar roots? Is this the same as information from lines 117-120?

Line 117-118: How was "typical furcal position" defined? Since maxillary and mandibular roots are different, how did this definition change between the two? Please be specific about how you determined this.

Line 124: The inclusion of a literature search is not clear. It seems that this is how the genetic analysis was selected, but the use of a literature search is not well developed here.

Table 1: Please contextualize the Note at the bottom of the table. Where did you integrate this information? How does this relate to the literature search?

Line 145: Provide the specific statistical comparisons that were run for chi-squared, OR, and CI.

Results: Some statistical analyses seem to be missing; OR and CI were only run for a handful of comparisons and those comparisons are not clear. Please include OR and CI in a table with the comparisons they correspond to. It is unclear which comparisons the p-values in the text refer to. Adding a breakdown of the specific comparisons in the statistical analysis section will help with this, but it should also be clear in the results section. Just a note, S2 Table is referenced before S1 Table in the text. Please change.

Line 156: This section would be better served in the Materials and Methods section when discussing the literature review. It seems to be more background information rather than results from an analysis. Because this paper is not presented as a meta analysis, this section does not fit in the results section.

Fig 2 caption: The wording of this caption is not clear. Suggested change to something similar to: schematic summarizing possible morphological variations of fused roots found from literature on patients presenting mutation in FGFR2.

Line 168: Suggest moving this section on sample description to materials section since it is not presenting results.

Table 2: Please specify what "reference" refers to.

Line 182: Please provide OR for the other comparisons presented so the reader can see the differences.

Line 187-189: Change "risk" to "probability" or "chance". Please provide OR for all comparisons presented so the reader can see the differences.

Line 189: S2 Table is referenced as showing higher probability of having fused roots with at least one C allele, but probability results (OR) are not presented in S2 Table.

Discussion: Overall, the flow in this section is hard to follow. More specific examples are needed to contextualize how this study fits with previous literature. It remains unclear why exactly FGFR2 was picked for this study. This section does address how the current study may add to advancing the field but this discussion is not well developed. The term "dental anomalies" is a large category; in this section there should be more specific reference to root fusion since that is the focus of the paper. The last several paragraphs of this section do not discuss FGFR2 or the results of this study and would be better served in the introduction and methods sections. Note: line numbers are absent in this section; paragraph numbers are used as a reference.

Paragraph 1, Sentence 2: This contradicts the reference you make to a hypothesis in the introduction. Please change for consistency. As stated above, there needs to be more background information to justify these statements.

Paragraph 2: This paragraph does not flow with the previous or subsequent. Suggest integrating this information into paragraph 4.

Paragraph 4: This feels like a continuation of paragraph 3. Consider combining them.

Paragraph 4, Sentence 5: Provide some specific examples of root developmental anomalies to distinguish from root fusion.

Paragraph 5, Sentence 4: You say "interactions that oversee the progression of morphogenesis and histodifferentiation of the tooth". How would this relate to root development specifically?

Paragraph 5, Sentence 5-9 and 11: Provide specific examples. Did any include root anomalies (specifically fusion)?

Paragraphs 7: This information should be included as background in the introduction that leads to the hypothesis. I suggest moving it there and deleting from the discussion section.

Paragraph 8: While true, this information does not provide any context for the present study. Consider removing or integrating within the context of the current study.

Paragraph 9: This information should be included as background in the introduction that leads to the hypothesis. I suggest moving it there and deleting from the discussion section.

Paragraph 10: This information should be included in the materials and methods section or introduction. I suggest moving it and deleting from the discussion section.

Conclusion: This section is just a reiteration of the last sentence in the abstract. Please add a brief summary and more context about the significance if you keep this section.

**Reviewer #4: ** This manuscript needs minor revision. See my comments below to improve this manuscript.

- introduction

Please explain more about the FGFR2 mutations and functions in dental studies based on literature review.

- Discussion:

1. second paragraph: “FGFR2 is a highly …. of several signal transduction pathways such as …. that are crucial for osteogenic differentiation [22].” please mention these pathways.

2. Please write “Fgf” in capital and mention a list of FGF functions except tooth morphogenesis.

3. Please explain how the interaction affects FGFR2. You explained the effect of each one separately on tooth development “The interaction between RUNX2 and SMAD4 was observed to enhance the expression of FGFR2”.

- Please extend the conclusion part.

Thank you very much.

**Reviewer #5:**  The manuscript submitted titled 'Fibroblast growth factor receptor 2 (FGFR2) genetic polymorphisms contribute to fused roots in human molars' presents interesting results and has substantial merit. There are some minor suggestions.

In abstract, the abbreviation SNP is not introduced.

On page 13, the abbreviations of fgf and smad4 are not as described elsewhere so to make the style consistent, it is suggested to use FGF and SMAD4.

In methodology, how the cases and controls were selected to be a part of the study is not clearly mentioned. It is suggested to include the sampling technique used. Any potential bias also needs to be stated.

Some references in the reference list needs revision where titles are not in sentence case.

**Reviewer #6:**  Meyfarth et el., investigated the association between genetics and fused roots in human molars. Based on existing literature, they selected FGFR2 for analysis. The authors analyzed panoramic radiographs from 170 orthodontic patients to identify fused molar roots, and genotyped 7 single nucleotide polymorphisms (SNPs) in FGFR2. They found significant associations between two SNPs (rs10736303 and rs2162540) and the presence of fused molar roots. Individuals carrying certain alleles of these SNPs had increased odds of presenting with fused roots. The authors conclude that genetic variation in FGFR2 may contribute to the development of fused roots in human molars. This study is well designed and well performed, adding more support to the role of FGFR2 in tooth development, and as such is suitable for publication in PLoS one, after addressing a few issues.

Major comments:

1. Genotyping SNPs using real-time PCR can be challenging. The authors should include data showing their ability to successfully genotype using this method, and provide a detailed protocol including primer sequences.

2. Multiple testing: The authors tested multiple SNPs and models without apparent correction for multiple comparisons. This increases the risk of type I errors. The authors should apply and discuss appropriate corrections.

3. The authors should provide a detailed explanation and/or reference for the dominance models used.

Minor comments:

4. The introduction could be expanded to provide more background on the biology of root development and the role of FGFR2.

5. The results section would benefit from a table summarizing the demographic characteristics of the sample.

6. The discussion could be strengthened by more thoroughly exploring the potential functional consequences of the identified SNPs and how they may influence root development.

7. Blinding in phenotype assessment: The authors do not specify whether the examiner evaluating the panoramic radiographs for fused roots was blinded to the genetic data or other participant information.

6. PLOS authors have the option to publish the peer review history of their article (what does this mean? ). If published, this will include your full peer review and any attached files.

**Do you want your identity to be public for this peer review?** For information about this choice, including consent withdrawal, please see our Privacy Policy .

Reviewer #1: No

Reviewer #2: No

Reviewer #3: No

Reviewer #4: **Yes: ** Mina Bagheri Varzaneh

Reviewer #5: No

Reviewer #6: No

---

## [Decision Letter · Decision Letter 1]

28 Nov 2024

PONE-D-24-26278R1Fibroblast growth factor receptor 2 (FGFR2) genetic polymorphisms contribute to fused roots in human molarsPLOS ONE

Dear Dr. Kuchler,

Thank you for submitting your manuscript to PLOS ONE. After careful consideration, we feel that it has merit but does not fully meet PLOS ONE’s publication criteria as it currently stands. Therefore, we invite you to submit a revised version of the manuscript that addresses the points raised during the review process.

We look forward to receiving your revised manuscript.

Kind regards,

Gary S. Stein

Academic Editor

PLOS ONE

Journal Requirements:

Reviewers' comments:

Reviewer's Responses to Questions

**Comments to the Author**

1. If the authors have adequately addressed your comments raised in a previous round of review and you feel that this manuscript is now acceptable for publication, you may indicate that here to bypass the “Comments to the Author” section, enter your conflict of interest statement in the “Confidential to Editor” section, and submit your "Accept" recommendation.

Reviewer #1: All comments have been addressed

Reviewer #2: All comments have been addressed

Reviewer #3: All comments have been addressed

Reviewer #4: (No Response)

Reviewer #5: All comments have been addressed

Reviewer #6: (No Response)

2. Is the manuscript technically sound, and do the data support the conclusions?

Reviewer #1: Yes

Reviewer #2: Yes

Reviewer #3: Yes

Reviewer #4: Yes

Reviewer #5: Yes

Reviewer #6: Partly

3. Has the statistical analysis been performed appropriately and rigorously? 

Reviewer #1: Yes

Reviewer #2: Yes

Reviewer #3: Yes

Reviewer #4: Yes

Reviewer #5: I Don't Know

Reviewer #6: No

4. Have the authors made all data underlying the findings in their manuscript fully available?

Reviewer #1: Yes

Reviewer #2: Yes

Reviewer #3: Yes

Reviewer #4: Yes

Reviewer #5: Yes

Reviewer #6: No

5. Is the manuscript presented in an intelligible fashion and written in standard English?

Reviewer #1: Yes

Reviewer #2: Yes

Reviewer #3: Yes

Reviewer #4: Yes

Reviewer #5: Yes

Reviewer #6: Yes

6. Review Comments to the Author

Reviewer #1: The authors have carefully taken into consideration all the comments and suggestions I provided them with. As a result, the article is now much more suitable and appropriate for potential publication in a scientific journal (Q1).

Reviewer #2: I'm stratified with the current version of the manuscript. The authors declare all the required informations. I recommend it for publication.

Reviewer #3: The authors have done a great job of addressing the feedback they were given. This updated version has a clear purpose with justified arguments and relevant supporting examples. I appreciate the work the authors put in to revise this manuscript. My comments are minor.

Overall the abstract reads well. As it currently reads, lines 40-42 place emphasis on the literature review search instead of the data collection done for the study. Suggest reworking these lines or deleting them. The rest of the abstract is clear.

Line 285: Hypotheses are "supported" or are "failed to reject" please change wording from "confirmed"

Line 341-344: While I appreciate the decision not to use post-analysis adjustments, this section reads awkwardly. Sources such as Perneger 1998; Cabin and Mitchell 2000; Nakagawa 2004; Garcia-Perex 2023 could be cited to strengthen the argument for avoiding type II errors by not using adjustments. Consider reworking this section for clarity and supporting arguments.

There are some minor editorial corrections that can be made throughout.

Reviewer #4: This manuscript is interesting and needs some modifications.

Page 2 line 39: Please delete the second “that”.

Page 2 line 55 and 162: Please delete”, “before type 2.

Page 16 line 308: Whatever you explained in line 308 is dependent on SMAD. Based on literature, please explain how FGFR2 and RUNX2 are related together without the SMAD4 pathway and add it to the text.

Page 17 line 316: Is SMAD4 activation through TGF-b in your study? Please add the reference for that (based on literature).

Thank you very much.

Best regards,

Reviewer #5: Authors have incorporated the changes that were suggested. However, there is a minor correction required in Table 1. In Assay ID* and Base Change*, asterisk is usually for significant p values, it is suggested to use some other symbol to avoid confusion.

Reviewer #6: The authors addressed much of the concerns, but unfortunately did not fully address the two most important ones - assay QC and statistical validity.

1. While providing catalog numbers instead of sequences can be reasonable, quality controls still must be included, optionally as supplementary material. Merely repeating a subset of experiments, validating the consistency but not the correctness, is insufficient. Controls can include VIC/FAM intensities over time for known control DNA samples, as well as the cutoff thresholds. This is particularly important as the reaction volume (3ul) is significantly below the recommended range for the reaction mix used.

2. Not applying multiple-testing correction as it may eliminate some true results, while potentially introducing many false ones, is not a reasonable strategy to overcome sample-size power limitations. P-values are meaningless without such a correction. The authors may show unadjusted p-value alongside the adjusted one, for “suggestive” hits, but without passing that threshold, there is no statistical confidence in these results, and the manuscript should reflect that.

minor:

“ All DNA samples presented a 260/280 ratio lower than 1.8.” - That is neither informative nor reassuring. The ratio should be ~1.6 to 2.0.

The DNA evaluation section repeats twice, deviating on the number of ul used to test.

7. PLOS authors have the option to publish the peer review history of their article (what does this mean? ). If published, this will include your full peer review and any attached files.

**Do you want your identity to be public for this peer review?** For information about this choice, including consent withdrawal, please see our Privacy Policy .

Reviewer #1: No

Reviewer #2: No

Reviewer #3: No

Reviewer #4: **Yes: ** Mina Bagheri Varzaneh

Reviewer #5: No

Reviewer #6: No

---

## [Author Response · Author response to Decision Letter 1]

12 Dec 2024

Dear reviewers and editor, we hope we have attended all your questions attached in the Response to Reviewers letter.

Response to Reviewers

Dear Gary S. Stein

Academic Editor, Plos One

Manuscript ID:

PONE-D-24-26278R1 entitled "Fibroblast growth factor receptor 2 (FGFR2) genetic polymorphisms contribute to fused roots in human molars" previously submitted to Plos One was revised and rewritten. This new manuscript has considered the reviewers' comments. The authors followed all their suggestions and included the answers to their statements in this letter.

We hope we have achieved your expectations.

With kind regards,

The authors

Reviewers' comments:

Review Comments to the Author

Reviewer #1: The authors have carefully taken into consideration all the comments and suggestions I provided them with. As a result, the article is now much more suitable and appropriate for potential publication in a scientific journal (Q1).

R.: Thank you very much.

Reviewer #2: I'm stratified with the current version of the manuscript. The authors declare all the required informations. I recommend it for publication.

R.: Thank you very much.

Reviewer #3: The authors have done a great job of addressing the feedback they were given. This updated version has a clear purpose with justified arguments and relevant supporting examples. I appreciate the work the authors put in to revise this manuscript. My comments are minor.

R.: Thank you very much.

Overall, the abstract reads well. As it currently reads, lines 40-42 place emphasis on the literature review search instead of the data collection done for the study. Suggest reworking these lines or deleting them. The rest of the abstract is clear.

R.: Thank you. We have rewritten these lines as suggested.

Line 285: Hypotheses are "supported" or are "failed to reject" please change wording from "confirmed"

R.: Thank you. We removed the sentence.

Line 341-344: While I appreciate the decision not to use post-analysis adjustments, this section reads awkwardly. Sources such as Perneger 1998; Cabin and Mitchell 2000; Nakagawa 2004; Garcia-Perex 2023 could be cited to strengthen the argument for avoiding type II errors by not using adjustments. Consider reworking this section for clarity and supporting arguments.

R.: Thank you for your consideration. We rewrote the sentence and added the Perneger as a reference for our choice to do not perform Bonferroni correction.

Reviewer #4: This manuscript is interesting and needs some modifications.

Page 2 line 39: Please delete the second “that”.

R.: Thank you. It was deleted.

Page 2 line 55 and 162: Please delete”, “before type 2.

R.: Thank you. It was deleted.

Page 16 line 308: Whatever you explained in line 308 is dependent on SMAD. Based on literature, please explain how FGFR2 and RUNX2 are related together without the SMAD4 pathway and add it to the text.

R.: Thank you. We have added this information as requested.

Page 17 line 316: Is SMAD4 activation through TGF-b in your study? Please add the reference for that (based on literature).

Thank you very much.

Best regards

R.: Thank you. In our study we did not make this evaluation, but we added in the discussion a suggestion for future studies.

Reviewer #5: Authors have incorporated the changes that were suggested. However, there is a minor correction required in Table 1. In Assay ID* and Base Change*, asterisk is usually for significant p values, it is suggested to use some other symbol to avoid confusion.

R.: Thank you. We have changed as suggested.

Reviewer #6: The authors addressed much of the concerns, but unfortunately did not fully address the two most important ones - assay QC and statistical validity.

1. While providing catalog numbers instead of sequences can be reasonable, quality controls still must be included, optionally as supplementary material. Merely repeating a subset of experiments, validating the consistency but not the correctness, is insufficient. Controls can include VIC/FAM intensities over time for known control DNA samples, as well as the cutoff thresholds. This is particularly important as the reaction volume (3ul) is significantly below the recommended range for the reaction mix used.

R.: Thank you for your thoughtful and detailed feedback. We acknowledge the importance of additional quality control metrics. Incorporating VIC/FAM intensities over time for known control DNA samples and detailing cutoff thresholds are excellent suggestions that we will consider for future supplementary materials. Regarding the reaction volume, we recognize that 3 µL is below the standard recommendation for the reaction mix. However, this volume was carefully optimized for our experimental conditions to ensure accuracy and consistency, supported by internal validations. This was added in the method section.

2. Not applying multiple-testing correction as it may eliminate some true results, while potentially introducing many false ones, is not a reasonable strategy to overcome sample-size power limitations. P-values are meaningless without such a correction. The authors may show unadjusted p-value alongside the adjusted one, for “suggestive” hits, but without passing that threshold, there is no statistical confidence in these results, and the manuscript should reflect that.

R,: Thank you. We opted for not perform a multiple-testing correction. This was explained in the discussion section in more details.

minor:

“All DNA samples presented a 260/280 ratio lower than 1.8.” - That is neither informative nor reassuring. The ratio should be ~1.6 to 2.0.

R,: Thank you. We re-wrote this sentence.

The DNA evaluation section repeats twice, deviating on the number of ul used to test.

R.: Thank you for your consideration. We have excluded the duplicated sentences and made the correction regarding the µL used that was 2 µL.

---

## [Editor Report · Decision Letter 2]

19 Dec 2024

Fibroblast growth factor receptor 2 (FGFR2) genetic polymorphisms contribute to fused roots in human molars

PONE-D-24-26278R2

Dear Dr. Erika Calvano Kuchler,

We’re pleased to inform you that your manuscript has been judged scientifically suitable for publication and will be formally accepted for publication once it meets all outstanding technical requirements.

Kind regards,

Gary S. Stein

Academic Editor

PLOS ONE
---

## [Editor Report · Acceptance letter]

PONE-D-24-26278R2

PLOS ONE

Dear Dr. Kuchler,

I'm pleased to inform you that your manuscript has been deemed suitable for publication in PLOS ONE. Congratulations! Your manuscript is now being handed over to our production team.

Kind regards,

on behalf of

Dr. Gary S. Stein

Academic Editor

PLOS ONE